# Differential Immunomodulatory Effect of Graphene Oxide and Vanillin-Functionalized Graphene Oxide Nanoparticles in Human Acute Monocytic Leukemia Cell Line (THP-1)

**DOI:** 10.3390/ijms20020247

**Published:** 2019-01-10

**Authors:** Sangiliyandi Gurunathan, Min-Hee Kang, Muniyandi Jeyaraj, Jin-Hoi Kim

**Affiliations:** Department of Stem Cell and Regenerative Biotechnology, Konkuk University, Seoul 05029, Korea; pocachippo@gmail.com (M.-H.K.); muniyandij@yahoo.com (M.J.)

**Keywords:** human acute monocytic leukemia cell, graphene, immunotoxicity, mitochondria, oxidative stress, apoptosis, cytokines

## Abstract

Graphene and its derivatives are emerging as attractive materials for biomedical applications, including antibacterial, gene delivery, contrast imaging, and anticancer therapy applications. It is of fundamental importance to study the cytotoxicity and biocompatibility of these materials as well as how they interact with the immune system. The present study was conducted to assess the immunotoxicity of graphene oxide (GO) and vanillin-functionalized GO (V-rGO) on THP-1 cells, a human acute monocytic leukemia cell line. The synthesized GO and V-rGO were characterized by using various analytical techniques. Various concentrations of GO and V-rGO showed toxic effects on THP-1 cells such as the loss of cell viability and proliferation in a dose-dependent manner. Cytotoxicity was further demonstrated as an increased level of lactate dehydrogenase (LDH), loss of mitochondrial membrane potential (MMP), decreased level of ATP content, and cell death. Increased levels of reactive oxygen species (ROS) and lipid peroxidation caused redox imbalance in THP-1 cells, leading to increased levels of malondialdehyde (MDA) and decreased levels of anti-oxidants such as glutathione (GSH), glutathione peroxidase (GPX), super oxide dismutase (SOD), and catalase (CAT). Increased generation of ROS and reduced MMP with simultaneous increases in the expression of pro-apoptotic genes and downregulation of anti-apoptotic genes suggest that the mitochondria-mediated pathway is involved in GO and V-rGO-induced apoptosis. Apoptosis was induced consistently with the significant DNA damage caused by increased levels of 8-oxo-dG and upregulation of various key DNA-regulating genes in THP-1 cells, indicating that GO and V-rGO induce cell death through oxidative stress. As a result of these events, GO and V-rGO stimulated the secretion of various cytokines and chemokines, indicating that the graphene materials induced potent inflammatory responses to THP-1 cells. The harshness of V-rGO in all assays tested occurred because of better charge transfer, various carbon to oxygen ratios, and chemical compositions in the rGO. Overall, these findings suggest that it is essential to better understand the parameters governing GO and functionalized GO in immunotoxicity and inflammation. Rational design of safe GO-based formulations for various applications, including nanomedicine, may result in the development of risk management methods for people exposed to graphene and graphene family materials, as these nanoparticles can be used as delivery agents in various biomedical applications.

## 1. Introduction

Graphene and graphene family materials (GFM), including graphene oxide (GO), reduced/functionalized graphene oxide (rGO), graphene quantum dots, graphene nanoribbons, three-dimensional graphene foam, and graphene nanopores, show immense potential for variety of biomedical applications, such as antibacterial, anticancer, drug delivery, bio-sensing, and bio-imaging applications, because of their excellent physical, chemical, mechanical, and biological properties, large surface area, ease of surface functionalization, and significant colloidal stability in aqueous media compared to pristine graphene [1,2,3,4]. GO and rGO have different physical and chemical properties such as solubility and dispensability, lateral dimension, sheet size, and oxidation and reduction degrees. The type of reducing agent used for reduction can influence the cellular uptake, toxicity, and biodegradation of these materials [5,6,7]. Graphene can be synthesized using various methods, such as physical mechanical cleavage or the scotch tape method [8], arc discharge [9], chemical methods [10], chemical vapor deposition [10], chemical oxidation [11], and longitudinal unzipping of carbon nanotubes [12]. The synthesis method determines the size, morphology, solubility, toxicity, and biocompatibility of graphene. However, nanosized graphene materials must be synthesized for biomedical applications, such as to cause either toxicity or biocompatibility. Gurunathan and coworkers demonstrated the different biological properties of GO and rGO in different types of bacteria as well as cancer and non-cancer cells [13,14,15,16,17]. Studies by Chatterjee et al., [18] demonstrated the toxic effects of GO and rGO with different hydrophilicity/hydrophobicity levels on HepG2 cells. GO and rGO, which differ in size, showed different toxicities towards glioblastoma cell lines [19]. 

Because graphene family materials show potential for applications in humans, studies examining the inevitable cytotoxic effects of these nanoparticles are necessary. Although the effects of GFM were reported in cancer, few studies have evaluated these effects on the immune system. Recent studies revealed that GO nanomaterials induce affect immune cells during inflammation via complex mechanisms [20,21]. GO induces size-dependent hemolysis of red blood cells and pristine GO and functionalized GO shows differential toxic effects on primary human peripheral blood T lymphocytes [22,23]. The toxic effect is significantly associated with the dimension of the graphene sheets and interactions with cells. GO sheets showed strong “adsorption” to the plasma membrane of murine macrophage-like J774.A1 cells with low phagocytosis, while small GO sheets were more readily taken up by cells [24]. Additionally, large GO promoted pro-inflammatory polarization of macrophages both in vitro and in vivo. In contrast, Orecchioni et al., [25] reported that small GO sheets elicited more profound effects on human immune cells compared to large GO sheets. Nano-sized GO sheets were shown to induce membrane ruffling in a variety of different cell lines with concomitant shedding of membrane fragments [26]. Russier et al., [27] demonstrated that the size of GO sheets significantly impacted various cellular parameters such as cellular viability, reactive oxygen species (ROS) generation, and cellular activation.

Several studies have evaluated the effects of pristine graphene, GO, and rGO on various types of immune cells. Oxidation of graphene causes significant toxicity without inducing ROS generation in macrophages [28]. Graphene can induce cytotoxicity by decreasing the mitochondrial membrane potential and increasing ROS production in macrophages [29]. Sub-toxic concentrations of pristine graphene significantly stimulates the secretion of Th1/Th2 cytokines, including interleukin (IL)-1α, IL-6, IL-10, tumor necrosis factor (TNF)-α, and granulocyte macrophage-colony-stimulating factor (GM-CSF), as well as chemokines, such as monocyte chemoattractant protein (MCP)-1, macrophage inflammatory protein (MIP)-1α, MIP-1β, and RANTES, in primary murine macrophages and immortalized macrophages [30]. The size of the GO sheets not only causes different cellular responses, but also is involved in stimulating cytokine production. For instance, small-sized GO (100–500 nm) induced cellular activation and cytokine release compared to large GO sheets (1–10 μm) in peripheral blood mononuclear cells [25]. Specific functionalization of graphene modulate the impact of immune cells. For example, chemical modifications of GFM significantly affected the impact of GO on the immune system including by reducing toxicity. Feito et al., [20] demonstrated that GO functionalized by polyethylene glycol (PEG) reduced the proliferation of T- and B-lymphocytes and macrophages in a dose-dependent manner, whereas no significant changes were observed in cytokine secretion. Furthermore, GO decorated with PEG affected the cell cycle, apoptosis, and oxidative stress in Saos-2 osteoblasts, MC3T3-E1 preosteoblasts, and RAW-264.7 macrophages. Xu et al., [31] demonstrated that GO functionalized with PEG and different types of polyethyleneimine promoted maturation of dendritic cells and enhanced their cytokine secretion by activating multiple Toll-like receptor (TLR) pathways. Zhou et al., [32] developed a coating strategy using colloidal graphene incorporated by layer-by-layer assembly onto electrospun poly-ε-caprolactone, which successfully rendered neuronal scaffolds bio-functional and electroactive. The authors also demonstrated that functionalized graphene scaffolds reduced the inflammatory response and supported endogenous neuroblast migration [33]. Surface functionalization is essential for determining the toxicity or biocompatibility of a compound. For example, amine-modified graphene platelets showed potential biocompatibility after modification of their surface charges [34].

A recent study by Yan et al., [35] determined the differential mechanism of GO nanoplatelets (GONPs) and reduced GONPs (rGONPs) on THP-1 cells, in which GONPs induced the expression of anti-oxidative enzymes and inflammatory factors, whereas rGONPs showed a substantially higher cellular uptake rate and higher levels of nuclear factor-κB expression. This differential mechanism occurred because of their surface oxidation states. Understanding the interactions between eukaryotic cells and GFMs with cell membranes is key for developing graphene-enabled biomedical technologies and managing graphene health and safety issues. Most studies have focused on pristine graphene, sheets/platelets of graphene nanomaterials, and GO, whereas few studies have examined the effects of rGO exclusively on immune cells. Thus, we selected THP-1 cells as a model system to evaluate the impact of immunotoxicity of GO and V-rGO. Cultured THP-1 cells, a human acute monocytic leukemia cell line, are valuable as a model system for in vitro studies and translational research. In this study, we first investigated the differential toxicity effects of GO and V-rGO on THP-1 cells by evaluating cellular viability, proliferation, oxidative stress, mitochondrial membrane potential, ATP synthesis, antioxidants, apoptosis, DNA damage, and the inflammation response.

## 2. Results

### 2.1. Synthesis and Characterization of Graphene Oxide (GO) and Vanillin-Functionalized GO (V-rGO)

GO was prepared by Hummers’ Method [36], which involves oxidation of graphite with strong oxidants and acids for a long time. In this study, we prepared ultra-small GO by mild oxidation and two-step centrifugation. The UV–vis spectrum of synthesized GO particles showed two characteristic maximum absorption peaks at 231, which was attributed to a π–π* transition of aromatic C=C bonds, and a shoulder at 300 nm, corresponding to an n–π* transition of C=O bonds [37]. The hydrophilicity of the oxygenated graphene layers resulted in significant solubility and stability in water. The absorption peak for V-rGO was red-shifted to 262 nm (Figure 1A,B) because of the restoration of sp2 carbon atoms. This characteristic red-shift is considered as a monitoring tool for the reduction of GO [16]. In the X-ray diffraction (XRD) pattern of GO, the strong and sharp peak at 2θ = 11.8° corresponded to an interlayer distance of 7.6 Å (d002). V-rGO showed a broad peak centered at 2θ = 25.9°, corresponding to interlayer distances of 3.53 Å (Figure 1C,D). These XRD results confirmed the exfoliation and reduction of GO and removal of intercalated water molecules and oxide groups. After reduction by vanillin, a broad peak was observed at 2θ = 25.9°, indicating the presence of stacked graphene layers. This indicates that a few layers graphene had formed [38]. Collectively, these findings agree with those of previous studies.

The reduction of GO by vanillin was confirmed by Fourier-transform infrared spectroscopy (FT–IR). As shown in Figure 1E (top panel), the most prominent peaks were observed in the spectrum of GO at 3315, indicating the presence of O-H stretching vibration, whereas the V-rGO peak at 1640 cm^−1^ was attributed to C=O bonds in the carboxylic acid and carbonyl moieties. After further reduction of GO by vanillin, peaks were observed in the spectrum of GO at 3420 and 1730 cm^−1^, corresponding to C=O stretching vibrations of COOH groups, which were attributed to C=O bonds in the carboxylic acid and carbonyl moieties, respectively, and at 1170 and 860 cm^−1^, corresponding to the asymmetric stretching and bending of –C–O–C– (epoxy) vibrations, respectively. The strong peaks at 1170, 1060, 860, and 570 cm^−1^ were due to C–OH stretching, C–O stretching, C–H in plane bending, and possibly C–H out-of-plane wagging vibrations, respectively. Interestingly, after the reduction process, the stretching vibration of the C=O and OH groups disappeared, and the number of other oxygen-containing functional groups in GO was significantly decreased (Figure 1F). Further reduction in the height of the C=O peak at 1730 cm^−1^ compared to that of GO strongly supports the idea that GO was functionalized by vanillin.

The nature of prepared GO and V-rGO was verified by dynamic light scattering (DLS) in the solution without the drying step. The sizes and uniformly sized dispersions of the GO and V-rGO particles were evaluated directly in the solution. The particle size distributions of aqueous dispersions of GO and V-rGO were determined by DLS analysis. Size distribution analysis is an important factor in either toxicity or biocompatibility analysis in aqueous solution. Therefore, we determined the sizes of GO and V-rGO by DLS at a concentration of 250 μg/mL. The average hydrodynamic diameters of GO and V-rGO were 100 ± 10 and 110 ± 20 nm, respectively (Figure 1G,H). As shown in Figure 1I,J, the size of GO was between 50 and 300 nm, with an average size of 100 ± 10 nm. V-rGO was between 50 and 300 nm, with an average size of 110 ± 20 nm. Both GO and V-rGO showed significantly uniform size distributions. We found no significant size difference between GO and V-rGO. Similarly, Yan et al., [35] reported that the hydrodynamic sizes of GONPs and reduced GONPs ranged from 20 to 100 nm, with a maximum between 37 and 56 nm, respectively.

Previous studies reported that the size of graphene and reduced graphene sheets was 500 nm or greater. For instance, GO reduced with isocyanate and carboxyl showed average sizes of 560 ± 60 and 1110 nm, respectively. Previously, Gurunathan et al., [16] reported that the average hydrodynamic diameters of GO and resveratrol rGO were 325 ± 7 and 540 ± 20 nm, respectively. Carboxyl-functionalized graphene nanoplatelets increased in size from 385 nm (GO) to 1110 nm [39]. Muthoosamy et al., [40] reported that using mushroom extract and sonication, the average sizes of GO and rGO were 313 and 181 nm, respectively. The smaller sized rGO was formed mainly because 10 min of ultrasonication was conducted. 

Surface studies of the morphology, structure, and size of GO and V-rGO samples were conducted by scanning electron microscopy (SEM) and transmission electron microscopy (TEM) as shown Figure 1K,L. Oxidation of graphite into GO showed spliced graphite sheets as well as decreased sizes of GO sheets, resulting in unpredictable edges of GO monolayer sheets. Further, plate-like sheets of GO with irregular edges were observed. V-rGO exhibited a typical wrinkled structure that caused sheet folding. The differences between GO and V-rGO sheets were detected by scanning electron microscopy (SEM). The SEM images showed that the samples were formed of multiple two-dimensional sheets layered on top of each other and, importantly, that the GO films were more ordered and tightly packed, while the V-rGO sheets were more textured and had a wrinkled structure. TEM images of GO revealed thick flat flake layers, a rough surface, no crumpling, and an irregular shape with non-uniform particle sizes. In contrast, V-rGO had a thin flat flake and crumpled morphology, consistent with previous reports [41,42]. The wrinkled flake structure of V-rGO was similar to that of graphene sheets, which have a thin film morphology. The thin multi-layers and crumpled sheets were associated with the exfoliation process during elimination of intercalated oxygen and others functional groups between the layers by vanillin. Thus, exfoliation and reduction of GO resulted in the production of a disordered solid graphene structure in form of crumpled sheets. 

Raman spectroscopy can be used to characterize graphene and graphene-related materials; their structure and conductivity can be determined through this technique and the method can provide important information on defects, carbon sp2 vibrations, and stacking order [43]. Figure 1M shows the Raman spectra of GO with a peak at 1590 cm^−1^, which is related to the G band of the graphitic structure (sp2 carbon), and D band located at 1360 cm^−1^. As shown Figure 1N, the Raman spectrum of V-rGO displayed a significant peak for D and the G band positions. The Raman spectrum showed that the typical features of V-rGO were a G band at ~1595 cm^−1^ and G band at ~1355 cm^−1^. The D band was assigned to the breathing mode of the k-point phonons with A1g symmetry, whereas the G band introduced the E2g phonon of the carbon sp2 atoms and the G band is common for all sp2 carbon forms, arising from the C–C bond stretch. The G band in GO was shifted to a higher wave number because of the oxygenation of graphite, resulting in the formation of sp3 carbon atoms [16,44,45]. After reduction, ID/IG was increased to 1.93 (V-rGO) from 1.12 (GO), indicating the removal of most oxygen-containing functional groups. The ID/IG ratio in V-rGO was higher than that of graphene because the sp2 domains that newly formed during reduction were smaller than those of graphite [46]. Additionally, we observed a broad peak at 2700 cm^−1^ corresponding to the 2D band, further confirming the presence of a 2–3 layers of graphene sheets in V-rGO. The presence of a large D mode indicated that the graphene flakes were rather defective. Collectively, GO was converted into rGO by vanillin, which agrees with previously reported results [3,47].

### 2.2. Effect of GO and V-rGO on Cell Viability and Proliferation of THP-1 Cells

To assess toxicity vs biocompatibility and the differential modulatory effect of GO and V-rGO in THP-1 cells, THP-1 cells were exposed to various concentrations of GO and V-rGO (20–100 µg/mL) for 24 h and cell viability was determined by CCK-8 assay. The results suggested that GO and V-rGO had dose-dependent effects on THP-1 cells, whereas V-rGO showed significant effects compared to untreated cells and GO (Figure 2A) and the inhibitory effect of V-rGO was more apparent at increasing concentrations. Next, we examined the effect of GO and V-rGO on the proliferation of THP-1 cells. The cells were treated with various concentrations of GO and V-rGO (20–100 µg/mL) for 24 h and cell viability was determined by using BrdU. The results suggested that GO and V-rGO dose-dependently affected the proliferation of THP-1 cells, whereas V-rGO significantly affected proliferation compared to untreated and GO-treated cells (Figure 2B). The inhibitory effect of V-rGO was significantly higher with increasing concentrations. Yan et al., [35] observed that reduced GONPs (rGONPs) showed better effects than GONPs in THP-1 cells. Duan et al., [48] found that the loss of cell viability of single-layer graphene in A549 lung carcinoma cells and macrophage-like RAW264.7 cells occurred because of the formation of holes in the plasma membrane, leading to cell death. Gurunathan and co-workers demonstrated that GO reduced with different biomolecules showed stronger cytotoxicity compared to GO in various types of cancer and non-cancer cells [14,15,49]. The rate of cell death caused by GO is related to its hydrophilic nature, which can easily enter cells compared to hydrophobic rGO [18]. There are many contrasting reports describing the biological effects of GO and rGO in several bacteria [13,50,51] and mammalian cells [52,53]. Several studies suggested that the degree of toxicity depends on the degree of oxidation using mild or severe oxidizing agents as well as the availability of functional groups. Our findings agree with previous studies showing that rGO had higher toxicity than GO towards mammalian cells [14,52]. The highly toxic effect of rGO may be related to the better charge transfer between the bacteria and sharper edges of reduced GO during contact interaction [50]. Wu et al., (2018) compared the biocompatibility aspects between parental and differential reduced GOs towards macrophages using primary bone marrow derived macrophages (BMDMs) and J774A.1 cell line. The results revealed that RGOs were more toxic than pristine GO to both types of cells [54]. In this study, V-rGO was found to reduce cell viability and proliferation to a greater extent compared to GO because of the sharp edges of V-rGO. Therefore, for V-rGO and GO of approximately the same size, V-rGO was more toxic, supporting the role of functional groups in biological interactions. Overall, our cellular viability and proliferation data revealed different behaviors of GO and V-rGO based on various physicochemical properties, with small lateral dimensions and more functional groups inducing greater effects.

To confirm these results, we assessed the cytotoxicity of GO and V-rGO on THP-1 cells. The overall morphologies of THP-1 cells were observed in the presence and absence of GO and V-rGO. The cells were treated with (50 µg/mL) for 24 h and then observed by light microscopy. The morphologies of GO- and V-rGO-treated cells significantly differed from that of the control groups (Figure 2C). Unlike the control, cells cultured with GO and V-rGO were more rounded or had more of a crushed morphology compared to the untreated control cells. Compared to the control group, the cells were deformed in the GO and V-rGO exposure groups, and abnormalities in cell morphology and the loss of cell viability were increased by V-rGO exposure. 

### 2.3. GO and V-rGO Enhance Lactate Dehydrogenase (LDH) Leakage 

To measure the impact of GO and V-rGO on the membrane integrity of THP-1 cells, we measured LDH 24 h after exposure of THP-1 cells to GO and V-rGO. As expected, lactate dehydrogenase (LDH) leakage occurred in a dose-dependent manner from both GO- and V-rGO treated cells; however, the effect was significantly higher in V-rGO-treated cells (Figure 3A). Increased leakage was detected in the V-rGO-treated group, indicating that the membranes of V-rGO-exposed monocytes were severely compromised; disrupted membranes cannot maintain normal cellular functions. PEGylated GO nanosheets exhibited a strong immunological response and leakage of LDH from macrophages. GO and V-rGO disrupted cell membrane function and integrity, showing significant differences from the untreated group. Further, cell death due to membrane damage was confirmed in a Trypan blue exclusion assay, in which dead cells were stained in blue, while live cells were not stained. A significant difference was observed between the cell lines and with increasing GO and V-rGO concentrations (Figure 3B). V-rGO induced toxicity at a concentration of 20 μg/mL. Membrane damage was higher in v-rGO-treated cells than in GO-treated cells. Jaworski et al., [55] reported that graphene platelets altered the morphology, mortality, viability, and membrane integrity in U87 and U118 glioma cells. GO and graphene sheets exhibited dose-dependent effects on human erythrocytes and skin fibroblast cells. Graphene sheets induced significant cell death compared to GO by increasing ROS generation and membrane damage [22]. A recent study suggested that hydrated GO caused the highest cell death in THP-1 and BEAS-2B cells because it had the highest carbon radical density, which caused cell death via lipid peroxidation of the surface membrane and membrane lysis. Tabish et al., (2017) reported that low concentrations of rGO were able to induce late apoptosis and necrosis rather than early apoptotic events and also it was able to disintegrate the cellular membranes in a dose dependent manner in two different type of lung cancer cells such as A549 and SKMES-1 [56]. Collectively, the results of the Trypan blue exclusion viability assay were consistent with those of cell viability and LDH assays and rGO was more toxic, as reported for human mesenchymal stem cells and human ovarian cancer cells [49,57]. Moreover, V-rGO with few layers of graphene showed highest toxicity towards TSP-1 cells.

### 2.4. GO and V-rGO Induce Mitochondrial Dysfunctions

Mitochondria play a major role in regulating apoptotic cell death via several control points through an intrinsic pathway. The loss of mitochondrial membrane potential (MMP) is one mechanism causing cell death. To determine the effects of GO and V-rGO on MMP, THP-1 cells were treated with various concentrations of GO and V-rGO (20–100 µg/mL) for 24 h and then MMP was determined by using JC-1 dye. Increasing concentrations of GO and V-rGO caused significant loss of MMP compared to in the control group (Figure 4A). Particularly, V-rGO induced a greater loss of MMP. Lammel and coworkers reported that exposing cells to GO perturbed mitochondrial structure and function, which decreased the MMP and resulted in dysregulation of mitochondrial Ca^2+^ homeostasis [39,58]. Human primary endothelial cells exposed to few-layer graphene showed elevated cytoplasmic Ca^2+^ concentrations and mitochondrial membrane depolarization, and generation of a transition pore led to more severe mitochondrial damage and apoptosis initiation [59]. Similarly, several studies demonstrated that a variety of graphene family materials including nano-sized GO, pristine graphene [60], GO, and GO silver nanocomposite caused decreases in the MMP in neuroblastoma cancer cells [61]. Our findings agree with those of previous studies showing that GO and rGO treatments also caused mitochondrial damage and accelerated apoptosis by increasing oxidative stress [27,62]. Overall, these results suggest that the exposure of THP-1 cells to GO and V-rGO increases intracellular ROS, inducing the loss of MMP. 

Mitochondria are the major sources of intracellular ATP, and the MMP is crucial to ATP synthesis [38]. MMP loss can lead to decreased levels of ATP synthesis. Therefore, we determined the level of ATP in GO- and V-rGO-treated cells. THP-1 cells treated with GO and V-rGO showed reduced levels of ATP; particularly, V-rGO had significant effects on ATP synthesis compared to the control group (Figure 4B). Disturbances in the MMP led to disruption of the electron transfer chain and release of pro-apoptotic molecules from the mitochondria into the cytoplasm, eventually leading to decreased ATP synthesis. Graphene directly inhibits ETC complexes I–IV, resulting in depolarization of the mitochondria and consequent impairment of ATP production in human breast cancer cells [63]. Thus, the cellular ATP level was decreased possibly because of decreased MMP. 

### 2.5. Impact of GO and V-rGO on Reactive Oxygen Species (ROS) Generation and Lipid Peroxidation

Graphene-induced ROS can cause oxidative stress, mitochondrial damage, and initiation of lipid peroxidation. The generation of ROS is a common toxicity mechanism of graphene and graphene derivatives. The biological responses of cells is depends on physicochemical properties of graphene including ROS production (Tabish et al., 2018) [64]. To investigate this possibility, we performed an ROS assay to measure the oxidative stress generated by GO and V-rGO in THP-1 cells. To determine the effectiveness of ROS production by GO and V-rGO in THP-1 cells, the cells were treated with various concentrations (20–100 µg/mL) of GO and V-rGO for 24 h. The results showed that ROS generation in cells was concentration-dependent on GO and V-rGO exposure. After 24 h of exposure, GO and V-rGO at 100 μg/mL induced approximately a 1.5- and 3-fold increases in ROS in THP-1 cells compared to in untreated cells (Figure 5A). V-rGO induced an even higher level of ROS in THP-1 cells compared to GO, indicating the mechanism of toxicity for larger V-rGO compared to those of GO. 

Oxidative stress induced lipid peroxidation by GO and V-rGO was further supported by the measuring malondialdehyde (MDA) content. A highly significant induction in lipid peroxidation was observed in THP-1 cells exposed to GO and V-rGO after 24 h exposure. GO and V-rGO induced ROS generation in a dose-dependent manner in THP-1 cells (Figure 5B). The cytotoxicity, increased level of ROS formation, and increased level of MDA content was induced by rGO probably likely via the strong hydrophobic interactions of rGO with the cell membrane and eventual destruction by extremely sharp edges of the rGO [29,49,57]. The mechanism of elevated level of MDA content in V-rGO treated cells may be the strong hydrophobic interactions with the cell membrane and ROS formation. Similarly to the level of ROS generation, MDA content was increased in pristine graphene nanomaterial-treated Vero cells because of the physical interaction of rGO and the cell membranes [18,28]. Pristine graphene was also found to increase ROS an deplete MMP and mitochondria-mediated apoptosis in murine RAW 264.7 macrophages, an important effector cell of the innate immune system [28,29]. Therefore, the principal mechanism of GO- and rGO-induced toxicity is oxidative stress and lipid peroxidation in THP-1 cells. The toxicity of GO and reduced GO is still controversial. These diverse transformation processes result in significant variations in the structural properties, reactivity, dispersibility, stability and density of functional groups. Furthermore, functionalized GO showed less affinity for extracellular proteins relative to the affinity shown by the parent GO. This reduced affinity consequently led to stronger interactions with the cell membrane and higher cellular uptake. Carbon atoms in graphene are covalently bonded through three electrons from each atom to form a strong lattice, and the fourth valence electron on each carbon atom is delocalized. These half-filled orbitals of the carbon atoms form a bond that permits free motion of electrons. Therefore, these free electrons can be localized at the edge or defect sites of the graphene plane, which makes them more reactive and more likely to act as stable carbon-based free radicals. Another mechanism of higher toxicity of vanillin-functionalized GO is due to smaller in size compared to parental GO, which can enter into the cells very easily. Therefore, vanillin reduced GO increased cytotoxicity by causing more oxidative stress compared to parent GO. Thus, these findings imply that although GO undergoes similar reduction processes, the difference in the reduction pathways causes striking variations in toxicity. Moreover, the great reactivity and stability of vanillin-reduced graphene-based radicals might result in an even greater impact on toxicity.

### 2.6. Effect of GO and V-rGO on Antioxidants

ROS production and lipid peroxidation induced by GO and V-rGO affected cellular redox homeostasis and antioxidant levels, which were evaluated by measuring the expression levels of antioxidant proteins such as glutathione (GSH), glutathione peroxidase (GPx), super oxide dismutase (SOD), and catalase (CAT) (Figure 6). After 24 h of exposure of THP-1 cells to GO and V-rGO (50 µg/mL), the levels of the tested antioxidants were decreased. GO and V-rGO decreased total GSH by 25% and 50% and GPx by 40% and 60%, respectively. SOD and CAT levels was also decreased. Similarly GSH depletion was observed previously in GO- and rGO-treated human ovarian cancer cells [49]. During high levels of oxidative stress, cells adjust to maintain a balance between prooxidants and antioxidants. For instance, GSH (reduced form) becomes GSSG (oxidized GSH) through the action of glutathione reductase. Increased production of these enzymes results in decreased antioxidants such as GSH and CAT [65]. The results showed that GSH and GPx were significantly decreased in all treated cells. The positive control included silver nanoparticles (AgNPs)-treated THP-1 cells, which showed significant downregulation of both GSH and GPx. Anti-oxidative enzymes such as SOD and CAT are known to defend against endogenous ROS. GO and V-rGO treatment resulted in significant modulation in their expression levels (Figure 6). Similarly, human cervical cancer cells treated with GO silver nanocomposite exhibited decreased levels of SOD and CAT in human cervical cancer cells [66]. 

### 2.7. GO and V-rGO Induce Expression of Apoptotic Genes and Suppress Anti-Apoptotic Genes

The level of ROS and imbalance in antioxidants induced by GO and V-rGO may lead to simultaneous activation of apoptotic cell death by activation of pro-apoptotic gene expression and suppression of anti-apoptotic gene expression. To evaluate this hypothesis, THP-1 cells were treated with 50 µg/mL of GO and V-rGO for 24 h and then mRNA expression was examined. The results in Figure 7 show that mRNA levels of the *p53, p21, Bax, caspase-9, and caspase-3* genes were upregulated by 1–4-fold and Bcl-2 was downregulated. The prominent apoptotic features of GO and V-rGO were corroborated by increased levels of *p53, p21, and Bax* and concomitantly increased expression of *caspase-9 and caspase-3* as well as a corresponding decrease in Bcl-2 expression in THP-1 cells. The ultimate executioners of apoptosis, the caspases (-9 and -3), showed higher expression levels. V-rGO significantly induced all tested genes compared to GO. In contrast, GO treatments markedly activated *p53, p21, Bax caspase-9, and caspase-3,* but the effects were lower than those caused by V-rGO. Interestingly, V-rGO caused an increase in apoptosis with significant alterations in key apoptotic genes. *p53* is a vital regulator of apoptosis and plays an important role in various cellular processes, including transcription, DNA repair, genetic stability, senescence, cell-cycle control, and apoptosis [67], and it can modulate the mitochondrial outer membrane, thereby governing the *Bcl-2* family to trigger apoptosis [68]. Pristine graphene activated Bim and Bax via the mitogen-activated protein kinase and transforming growth factor-β signaling pathways in macrophages. Thus, caspase-3 and its downstream effector proteins, such as poly ADP ribose polymerase, were activated, resulting in apoptosis [29]. Cho et al. [69] reported that immunotoxicity of single- and multi-layered graphene oxides with or without pluronic F-127 in THP-1 cells. The results suggest that GOs exhibited dose- and size-dependent toxicity and SLGO (Single layered graphene oxide) induced ROS production to a lesser extent than multilayered graphene oxide (MLGO). Furthermore, SLGO induced necrosis and apoptosis to a lesser degree than MLGO [69]. Our results agree with those of previous reports showing that increased ROS generation leads to apoptosis and autophagy with increased expression levels of apoptotic and autophagic effectors, such as *caspase-3, caspase-9, Beclin-1, Bax, Bad, and LC3-I/II*, after exposure to graphene based materials GBMs [70,71].

### 2.8. Effect of GO and V-rGO on DNA Damage

The most important consequence of oxidative stress is damage to DNA. ROS generate various modified DNA bases. Among them, 8-oxo-7,8-dihydroguanine (8oxodG) is the most abundant and appears to play a major role in mutagenesis and carcinogenesis. Oxidative DNA damage refers to the oxidation of specific bases. 8-hydroxydeoxyguanosine is the most common marker of oxidative DNA damage [72]. To further explore the genotoxicity effect of GO and V-rGO, DNA damage was evaluated by measuring the level of 8-oxo-dG. The results showed that GO and V-rGO significantly increased 8-oxo-dG to 3 and 7 ng/mL, respectively (Figure 8). As a positive control, AgNPs increased 8-oxo-dG by more than GO and V-rGO. Carbon nanomaterials can activate the DNA damage response by enhancing the expression of p53, ATM, and Rad51 [73,74]. GO and V-rGO increase the level of ROS and may stimulate the expression of several DNA damage response genes such as OGG1, APEX1, CREB1, UNG, and POLB in THP-1 cells To assess the involvement of DNA damage-associated genes, the expression profiles of various genes were quantified following exposure of THP-1 cells to GO and V-rGO (50 µg/mL) for 24 h. The results suggested that GO and V-rGO significantly induced the expression of all tested genes. OGG1, APEX1, CREB1, UNG, and POLB were significantly upregulated by 1.5–3-fold. Lu et al., [75] reported that among the investigated genetic markers in HEK293T cells, base excision repair pathway genes (APEX1, OGG1, CREB1, UNG) were significantly upregulated upon exposure to a high GO dose (50 µg/mL); however, low exposure (5, 25 µg/mL) failed to induce significant genetic induction except that CREB1 was induced by 25 µg/mL GO. Previous studies suggested that graphene quantum dots trigger ROS generation [76,77], which play key roles in mediating DNA damage induced by physical and chemical agents [78]. Recently, Liu et al., [73] demonstrated that GO treatments at 10 and 100 mg/mL induced aberrant increases in ATM and Rad51 expression in cells, which are responsible for DNA damage. Collectively, GO and V-rGO may damage DNA. 

### 2.9. Effect of GO and V-rGO on Cytokine and Chemokine Production

GONPs are known to decrease cell viability and DNA damage, increase ROS production, and induce inflammatory factors. Therefore, we examined the immunomodulatory effect of GO and V-rGO on cytokine and chemokine release. To measure cytokine and chemokine responses, THP-1 cells were treated with GO and V-rGO for 24 h and LPS (1 µg/mL) was used as a positive control. The cytokine measurement data suggested that GO and V-rGO significantly induced the production of all cytokines and chemokines tested: IL1-β, TNF-α, GM-CSF, IL-6, IL-8, and MCP-1 (Figure 9). However, cytokine release was significantly higher in V-rGO-treated cells than in GO-treated cells. Similarly, Feito et al., [20] reported that RAW 264.7 cells exposed to PEG-functionalized GO upregulated TNF-α under both basal and LPS-stimulated conditions. The increased inflammatory responses were due to functionalization/reduction of GO using a variety of reducing agents. Our findings agree with those of Orecchioni et al., [25], who observed significant stimulation of the secretion of Th1/Th2 cytokines, such as IL-1a, IL-6, IL-10, TNF-α, and GM-CSF, as well as chemokines such as MCP-1, MIP-1α, MIP-1β, and RANTES in primary and immortalized murine macrophages. Zhi et al., (2013) investigated the immunotoxicity effect of graphene oxide (GO) on human immune cells such as dendritic cells (DCs), T lymphocytes and macrophages [79]. PVP-coated GO (PVP-GO) exhibited lower immunogenicity compared with pure GO on the aspect of inducing differentiation and maturation of dendritic cells (DCs). PVP-coated GO possesses good immunological biocompatibility and immunoenhancement effects. Lategan et al., (2018) also reported that RAW264.7 and human whole blood cell cultures exposed to GONPs showed significantly upregulated IL-6 [80]. Furthermore, through proteomic analysis, the authors observed activation of inflammatory cytokines and chemokines when whole blood cell cultures were exposed to 5 µg/mL GONPs. Notably, IL-1ra, MCP-1, MIP-1 α/β, and IL-8 were upregulated by GONPs [81]. Thus, stimulation of the secretion of cytokines and chemokines may be attributed to the interaction of GO and V-rGO with the Toll-like receptors. A recent study suggested that nanoparticles from photocopiers significantly elevated the levels of GM-CSF, IL-1β, IL-6, IL-8, IFNγ, MCP-1, TNF-α, and VEGF in THP-1 cells [82]. In contrast, GO did not trigger production of the classical pro-inflammatory Th1 cytokines TNF-α, IL-6, or IL-1β in macrophages. However, GO induced production of pro-inflammatory IL-8/CXCL8, an important chemotactic factor in neutrophils [83]. Our results clearly indicate that GO and V-rGO are cytotoxic at high concentrations and can stimulate inflammatory responses, eventually resulting in autoimmunity in individuals exposed to graphene family materials, graphene derivatives, and GONPs such as GO and rGO.

## 3. Materials and Methods 

### 3.1. Materials 

Penicillin-streptomycin solution, trypsin-EDTA solution, RPMI 1640 medium, and 1% antibiotic-anti-mycotic solution were obtained from Life Technologies/Gibco (Grand Island, NY, USA). Vanillin, fetal bovine serum, and the in vitro toxicology assay kit were purchased from Sigma-Aldrich (St. Louis, MO, USA). Graphite (Gt) powder, NaOH, KMnO4, NaNO_3_, anhydrous ethanol, 98% H_2_SO_4_, 36% HCl, 30% H_2_O_2_ aqueous solution, and all other chemicals were purchased from Sigma-Aldrich unless otherwise stated. The cell viability assay WST-8, ROS assay kit, 2′,7′-dichlorofluorescin diacetate, and lactate dehydrogenase (LDH) cytotoxicity detection kit were purchased from Dojindo (Kumamoto, Japan), Sigma, and Takara (Shiga, Japan), respectively. 

### 3.2. GO Synthesis, Reduction, and Characterization

Graphene sheets were synthesized by Hummer’s method with slight modification [17,84,85,86]. To obtain ultra-small uniform GO sheets, low-speed centrifugation at 5000 rpm was first used to remove the thick multilayer flakes until all visible particles were removed (15 min). The supernatant was further centrifuged at 8000 rpm for 10 min to obtain the GO sheets. To prepare size-controlled GO, 500 mg of as-synthesized GO powders was dissolved in 50 mL of deionized (DI) water and divided into two 20-mL vials. Each GO suspension (10 mg/mL in DI water) was subjected to ultrasonication using a probe type sonicator (Vibra-Cell VCX-500, 500 W, 20 kHz, Sonics & Materials, Inc., Newtown, CT, USA) at 25% amplitude for 10 and 60 min to yield GO sheets with different small sizes. An ice bath was used to avoid increasing the temperature during probe sonication.

The reduction of GO was accomplished as described previously [38]. Briefly, V-rGO was obtained from a reaction of vanillin with GO. In a typical reduction experiment, 10 mL vanillin (1 mM) was added to 90 mL of 1.0 mg/mL aqueous GO. The mixture was stirred at 40 °C for 6 h. Subsequently, using a magneto-stirrer heater, the rGO suspension was stirred at 400 rpm at 30°C for 30 min. A homogeneous V-rGO suspension was obtained without aggregation. The obtained material was washed with distilled water several times to remove the excess vanillin residue and re-dispersed in water by sonication. The suspension was centrifuged at 5000 rpm for 30 min. The final product was collected by vacuum filtration and vacuum-dried. GO and V-rGO were characterized as described previously [17].

Synthesis and characterization of AgNPs was carried out as described previously [16]. Synthesis of AgNPs was carried with vanillin dissolved in dimethyl sulfoxide (DMSO; 1 mM). AgNPs were synthesized by incubating 1 mM vanillin with 1 mM AgNO_3_ at 37 °C for 1 h. The color change from pale yellow to dark yellowish brown was attributed to the formation of AgNPs in the reaction mixture. 

### 3.3. Cell Viability Assay

THP-1 cells were cultured in RPMI 1640 medium (Gibco) supplemented with 10% (*v/v*) fetal bovine serum, 1% penicillin (*v/v*), and 1% streptomycin (*v/v*) (Sigma-Aldrich). The cells cultured to the logarithmic growth phase and mixed with various concentrations of GO and V-rGO (20, 40, 60, 80, and 100 µg/mL) for an additional 24 h, followed by cytotoxicity assays. 

### 3.4. BrdU Cell Proliferation Assay 

Cell proliferation was determined according to the manufacturer’s instructions (Roche, Basel, Switzerland). The cells were incubated with various concentrations of GO (20, 40, 60, 80, and 100 µg/mL) and V-rGO (20, 40, 60, 80, and 100 µg/mL) for 24 h; BrdU labeling solution was added to the culture medium 2 h before the end of incubation. Cells were fixed and the level of incorporated BrdU was determined using the Cell Proliferation enzyme-linked immunosorbent assay (ELISA) BrdU assay kit (Roche) following the manufacturer’s instructions. The proliferation activity of untreated cells at 0 h was considered as 100%. 

### 3.5. Measurement of Cytotoxicity

The membrane integrity of THP-1 cells was evaluated according to the manufacturer’s instructions (LDH Cytotoxicity Detection Kit; Takara). Briefly, the cells were exposed to various concentrations of GO and V-rGO for 24 h, 100 μL of each cell-free supernatant was transferred in triplicate into the wells of a 96-well plate, and 100 μL of the LDH reaction mixture was added to each well. After 3 h of incubation under standard conditions, the optical density of the color generated was determined at a wavelength of 490 nm using a microplate reader. 

### 3.6. Cell Mortality Assay

Cell mortality was evaluated using the trypan blue assay as described previously [38]. THP-1 cells were plated into 6-well plates (1 × 105 cells per well) and incubated for 24 h with various concentrations of GO or V-rGO (20–100 µg/mL). Cells cultured in medium without GO or V-rGO were used as controls. Twenty-four hours later, the cells were detached with 300 µL trypsin–EDTA solution and both adherent and suspended cells were collected. The mixture of the supernatant and detached cells was centrifuged at 1200 rpm for 5 min. The pellet was mixed with 700 µL of trypan blue solution and dispersed. After 5 min of staining, the cells were counted using a cytometer. The viable cells were unstained, whereas dead cells were stained blue. Three independent experiments were performed in triplicate. The mean and standard deviation were calculated. Cell proliferation was expressed as the percentage of viable cells.

### 3.7. Determination of ROS 

ROS were estimated as described previously [49,87]. The cells were seeded into 24-well plates at a density of 5 × 104 cells per well and cultured for 24 h. After washing twice with phosphate-buffered saline (PBS), fresh media containing various concentrations of GO and V-rGO were added and incubated for 24 h. The cells were then supplemented with 20 μM DCFH-DA, and incubation was continued for 30 min at 37 °C. The cells were rinsed with PBS, 2 mL of PBS was added to each well, and fluorescence intensity was determined using a spectrofluorometer (Gemini EM, Molecular Devices, Sunnyvale, CA, USA) at an excitation of 485 nm and emission of 530 nm. 

### 3.8. Determination of Malondialdehyde (MDA) 

The expression levels of oxidative and anti-oxidative stress markers were measured as described previously [87]. The malondialdehyde (MDA) level was measured according as described earlier [87]. 

### 3.9. Mitochondrial Membrane Potential (MMP)

The mitochondrial membrane potential (MMP) was measured according to the manufacturer’s instructions (Molecular Probes, Eugene, OR, USA) using a cationic fluorescent indicator JC-1 (Molecular Probes). THP-1 cells were incubated with 10 μM JC-1 at 37 °C for 15 min, washed with PBS, and resuspended in PBS, and then the fluorescence intensity was measured. MMP was expressed as the ratio of the fluorescence intensity of the JC-1 aggregates to that of the monomers. 

### 3.10. Measurement of ATP

The ATP level was measured according to the manufacturer’s instructions (Catalog Number MAK135, Sigma) in THP-1 cells treated with various concentrations of GO and V-rGO (20–100 µg/mL) for 24 h. The decreased levels of ATP indicated the cytotoxicity of the treated cells. 

### 3.11. Measurement of Anti-Oxidative Markers

Anti-oxidative stress markers such as GSH, GPx, SOD, and CAT were measured according to the manufacturer’s instructions. THP-1 cells were cultured in 75-cm^2^ culture flasks and exposed to different concentrations of GO and V-rGO (50 µg/mL) for 24 h. The cells were harvested in chilled PBS by scraping and washing twice with 1× PBS at 4 °C for 6 min at 1500 rpm. The cell pellet was sonicated at 15 W for 10 s (3 cycles) to obtain the cell lysate. The resulting supernatant was stored at −70 °C until analysis. 

### 3.12. Reverse Transcription-Quantitative Polymerase Chain Reaction (RT-qPCR)

Total RNA was extracted from the cells treated with GO and V-rGO for 24 h using the PicoPure RNA isolation kit (Arcturus Bioscience, Mountain View, CA, USA). Samples were prepared according to the manufacturer’s instructions. Real-time quantitative polymerase chain reaction (RT-qPCR) was conducted using a Vill7 (Applied Biosystems, Foster City, CA, USA) and SYBR Green as the double-stranded DNA-specific fluorescent dye (Applied Biosystems). Target gene expression levels were normalized to the expression of glyceraldehyde-3-phosphate dehydrogenase (GAPDH) expression, which was unaffected by treatment. The real-time qRT-PCR primer sets are shown in Appendix A. 

### 3.13. Measurement of Cytokines and Chemokines 

To monitor cytokine and chemokine production, THP-1 cells were incubated with GO, V-rGO (50 μg/mL), and LPS (1 μg/ mL) for 24 h and the cell culture samples were collected, centrifuged at 12,000 rpm for 5 min to remove cell debris, and stored at −80°C until analysis. Cytokine profiling was performed on the cell culture supernatants using the 13-plex human suspension cyto/chemokine assay kit (High-Sensitivity Human Cytokine Kit (Millipore, Billerica, MA, USA)) according to the manufacturer’s instructions. Cytokine concentrations were calculated using Upstate Beadview (Temecula, CA, USA) software.

### 3.14. Statistical Analysis

All assays were conducted in triplicate, and each experiment was repeated at least three times.

The results are presented as the means ± standard deviation. All experimental data were compared using Student’s *t*-test. A *P*-value less than 0.05 was considered statistically significant.

## 4. Conclusions

Graphene and graphene derivatives are increasingly utilized in biomedical applications because of their unique properties, including large surface area and mechanical, physical, chemical, and biological properties. Graphene family materials exhibited different degrees of toxicity depending on the size, oxidation, reduction, and solubility. However, studies have shown conflicting results regarding their biocompatibility versus toxicity and biosafety because of several factors. In this first study, we focused on the differential effect of GO and the biomolecule V-rGO with an average size of 100 nm by using the same starting material in THP-1 cells. We synthesized GO and V-rGO and characterized these materials using various analytical techniques. The results show that the biomolecule vanillin successfully functionalized GO and cellular assays showed that THP-1 cells treated with GO and V-rGO had dose-dependent effects on cell viability, proliferation, LDH leakage, MMP, intracellular ATP level, ROS generation, and lipid peroxidation. Furthermore, GO and V-rGO decreased antioxidant levels and increased the expression of pro-apoptotic genes and genes responsible for DNA damage. The distinct biological and molecular mechanisms of GO and V-rGO were attributed to their differential stimulatory effects on the secretion of various cytokines and chemokines in THP-1 cells. Among these two tested nanomaterials, V-rGO showed significant effects on immunotoxicity compared to GO because of the sharp edges, charge transfer, different carbon to oxygen ratio, chemical composition, and functional groups present on V-rGO. We developed a model for analyzing the molecular mechanism of the differential effects of graphene nanomaterials that can be utilized in more efficient and harmless applications, specifically in biomedical engineering and nanomedicine, and for predicting the modes-of-action of similar agents and the development of the safest material for the environmental and health care industries. Further mechanistic studies are required to better understand the trafficking patterns of these graphene nanoparticles inside cells and molecular mechanisms involved in immunotoxicity.

## Figures and Tables

**Figure 1 ijms-20-00247-f001:**
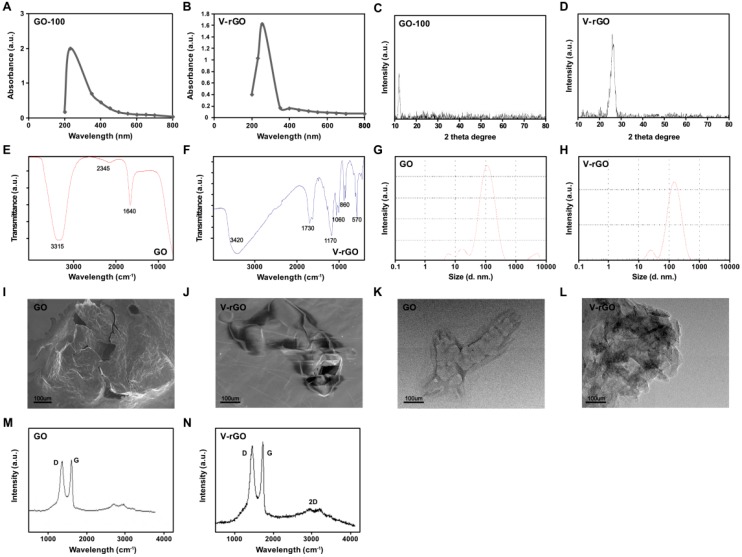
Synthesis and characterization of ultra-small graphene oxide. Ultraviolet-visible spectroscopy of graphene oxide (GO) (**A**) and vanillin- functionalized GO (V-rGO) (**B**). X-ray diffraction (XRD) images of GO (**C**) and V-rGO (**D**). FTIR images of GO (**E**) and V-rGO (**F**). Dynamic light-scattering (DLS) spectra of GO (**G**) and V-rGO (**H**). Scanning electron microscope (SEM) images of GO (**I**) and V-rGO-20 (**J**). Transmission electron microscope (TEM) images of GO (**K**) and V-rGO (**L**). Raman spectroscopy images of GO (**M**) and V-rGO (**N**). At least three independent experiments were performed for each sample and reproducible results were obtained. The data present the results of a representative experiment.

**Figure 2 ijms-20-00247-f002:**
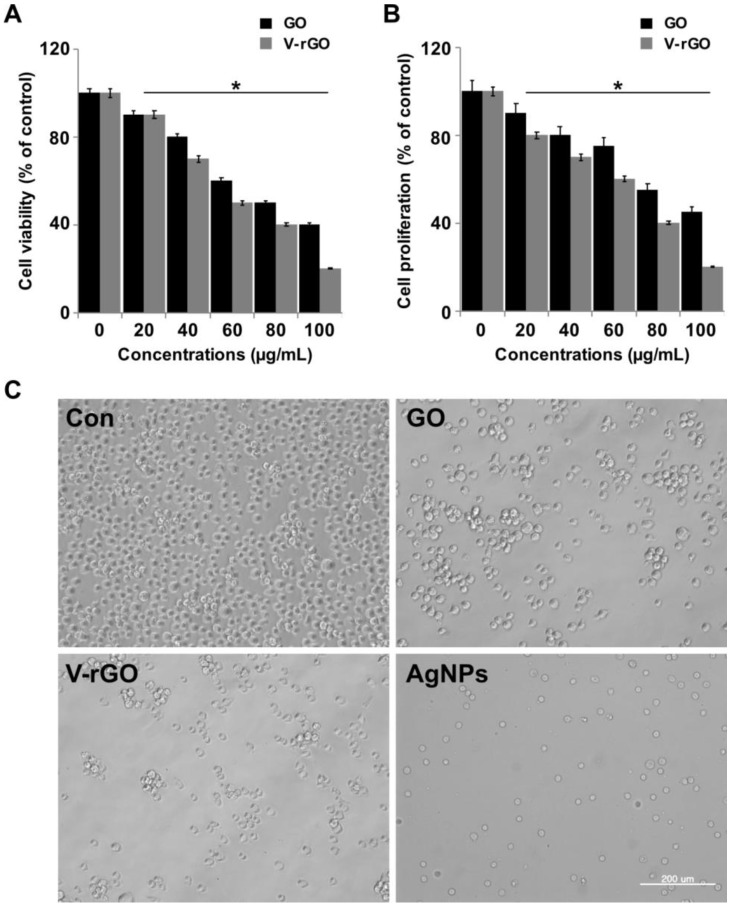
GO and V-rGO inhibits cell viability of THP-1 cells. (**A**) The viability of THP-1 cells was determined after 24-h exposure to different concentrations of GO (20–100 µg/mL) and V-rGO-20 (20–100 µg/mL) (**B**) Cell proliferation of THP-1 cells was determined using BrdU assay after 24-h exposure to different concentrations of GO (20–100 µg/mL) and V-rGO-(20–100 µg/mL). (**C**) Cell morphology was determined after 24-h exposure to different concentrations of GO (50 µg/mL) and V-rGO (50 µg/mL) using an optical microscope. The results are expressed as the mean ± standard deviation of three independent experiments. At least three independent experiments were performed for each sample. The treated groups showed significant differences from the control group by the Student’s *t*-test (* *P* < 0.05).

**Figure 3 ijms-20-00247-f003:**
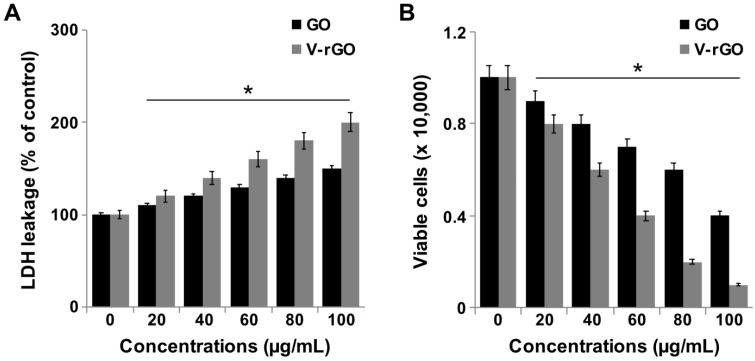
GO and V-rGO increase leakage of lactate dehydrogenase (LDH) and cell death.(**A**) THP-1 cells were treated with GO (20–100 µg/mL) and V-rGO (20–100 µg/mL) for 24 h, and LDH activity was measured at 490 nm using an LDH cytotoxicity kit. (**B**) Cell death was determined by a trypan blue assay after 24 h of exposure to GO (20–100 µg/mL) and V-rGO (20–100 µg/mL) for 24 h. Cell death was quantified as the ratio of living cells. At least three independent experiments were performed for each sample. The results are expressed as the mean ± standard deviation of three independent experiments. The treated groups showed significant differences from the control group by the Student’s *t*-test (* *P* < 0.05).

**Figure 4 ijms-20-00247-f004:**
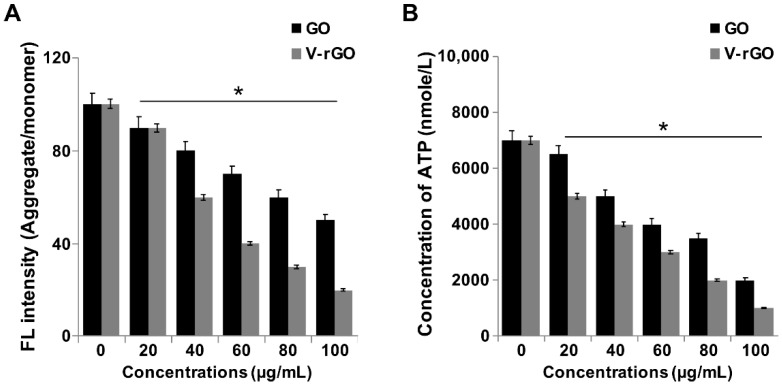
GO and V-rGO decrease mitochondrial membrane potential and ATP content (**A**) THP-1 cells were treated with GO (20–100 µg/mL), (**B**) V-rGO-20 (20–100 µg/mL) for 24 h, and the mitochondrial membrane potential (MMP) was determined using the cationic fluorescent indicator JC-1 (B). THP-1 cells were treated with GO (20–100 µg/mL), (**B**) V-rGO (20–100 µg/mL) for 24 h, and the intracellular ATP content was determined according to the manufacturer’s instructions (Sigma-Aldrich). The results are expressed as the mean ± standard deviation of three independent experiments. The treated groups showed significant differences from the control group by the Student’s *t*-test (* *P* < 0.05).

**Figure 5 ijms-20-00247-f005:**
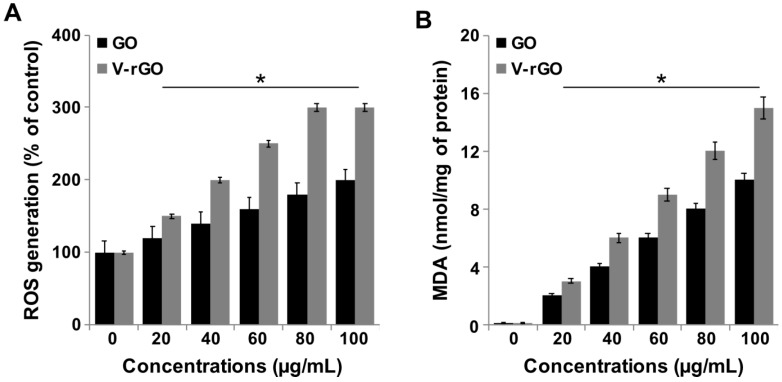
GO and V-rGO induce reactive oxygen species (ROS) generation and lipid peroxidation. (**A**) THP-1 cells were treated with GO (20–100 µg/mL) and V-rGO-20 (20–100 µg/mL) for 24 h and ROS level was measured using DCFH-DA. (**B**) THP-1 cells were treated with GO (20–100 µg/mL) and V-rGO (20–100 µg/mL) for 24 h and malondialdehyde (MDA) level was measured. After incubation, the cells were harvested and washed twice with ice-cold PBS. The cells were collected and disrupted by ultrasonication for 5 min on ice. The concentration of MDA was measured on a microplate reader at a wavelength of 530 nm. The results are expressed as the mean ± standard deviation of three independent experiments. The treated groups showed significant differences from the control group by Student’s *t*-test (* *P* < 0.05).

**Figure 6 ijms-20-00247-f006:**
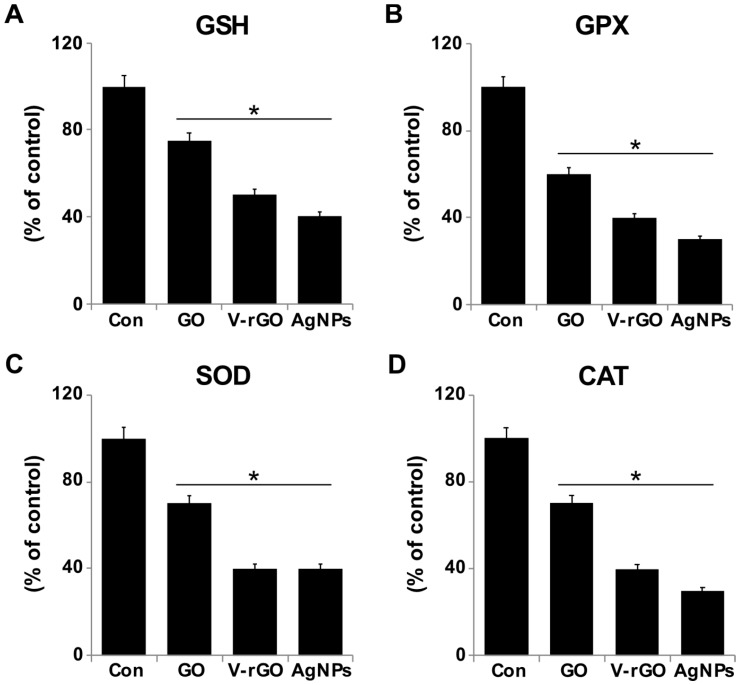
Effects of GO and V-rGO on anti-oxidants markers. THP-1 cells were treated with GO (50 µg/mL), V-rGO (50 µg/mL), and AgNPs (10 µg/mL) for 24 h. After incubation, the cells were harvested and washed twice with ice-cold phosphate buffered saline (PBS). The cells were collected and disrupted by ultrasonication for 5 min on ice. (**A**) Glutathione (GSH) concentration expressed as percentage of the control. (**B**) Glutathione peroxidase (GPx) concentration expressed as a percentage of the control. (**C**) Super oxide dismutase (SOD) expressed as percentage of the control. (**D**) Catalase (CAT) expressed as a percentage of the control. Results are expressed as the mean ± standard deviation of three independent experiments. There was a significant difference in treated cells compared to in untreated cells by Student’s *t*-test (**P* < 0.05).

**Figure 7 ijms-20-00247-f007:**
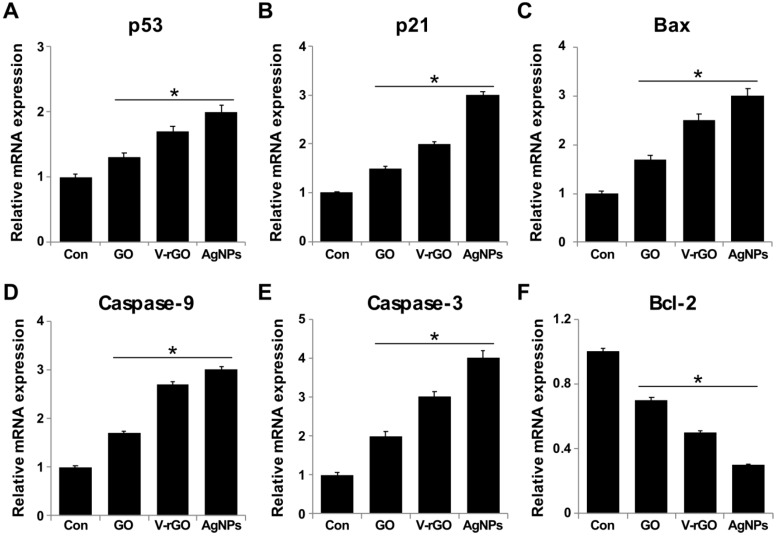
Effect of GO and V-rGO on expression of pro- and anti-apoptotic genes. THP-1 cells were treated with GO (50 µg/mL), V-rGO (50 µg/mL), and AgNPs (10 µg/mL) for 24 h. Relative mRNA expression of apoptotic and anti-apoptotic genes was analyzed by quantitative reverse-transcription polymerase chain reaction (PCR) in THP-1 cells treated for 24 h. After 24 h of treatment, expression was determined as fold-changes relative to glyceraldehyde-3-phosphate dehydrogenase (GAPDH) (**A**–**F**). Results are expressed as fold-changes. Results are expressed as the mean ± standard deviation of three independent experiments. There was a significant difference in treated cells compared to in untreated cells by Student’s *t*-test (* *P* < 0.05).

**Figure 8 ijms-20-00247-f008:**
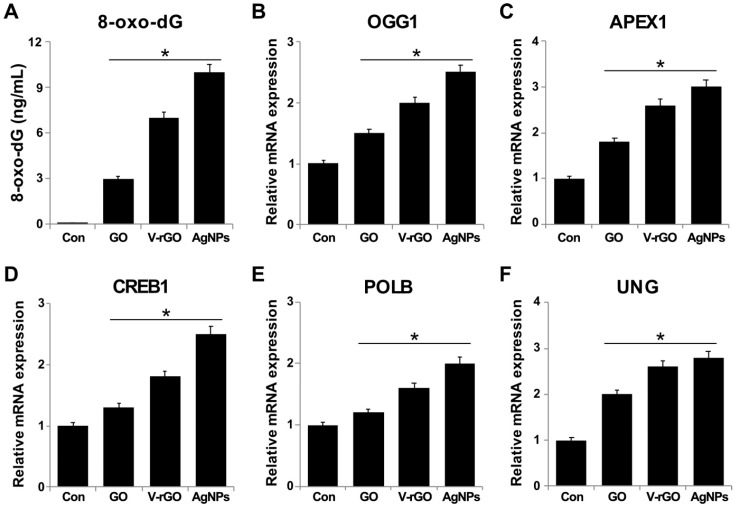
Effect of GO and V-rGO on DNA damage and expression of genes involved in DNA damage. THP-1 cells were treated with GO (50 µg/mL), V-rGO (50 µg/mL), and AgNPs (20 µg/mL) for 24 h. 8-oxo-dG was measured after 24 h of exposure of THP-1 cells. GO and V-rGO significantly increased oxidative DNA damage as evidenced by significant increases in 8 -oxo-dG expression. THP-1 cells were treated with GO (50 µg/mL), V-rGO-20 (50 µg/mL), and AgNPs (20 µg/mL) for 24 h. Relative mRNA expression of DNA damage genes was analyzed by quantitative reverse-transcription PCR in THP-1 cells. The expression was determined as fold-changes relative to GAPDH (**A**–**F**). Results are expressed as fold-changes. Results are expressed as the mean ± standard deviation of three independent experiments. There was a significant difference in treated cells compared to in untreated cells by Student’s *t*-test (* *P* < 0.05).

**Figure 9 ijms-20-00247-f009:**
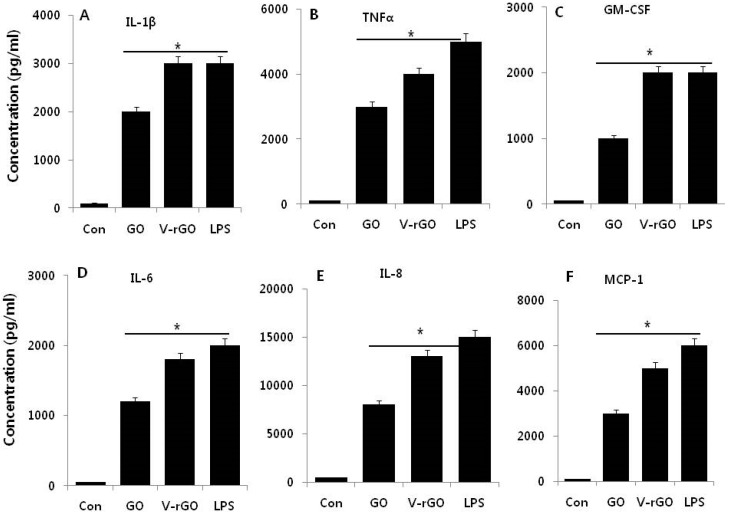
Effect of GO and V-rGO on level of cytokines and chemokines. THP-1 cells were treated with GO (50 µg/mL), V-rGO-20 (50 µg/mL), and AgNPs (10 µg/mL) for 24 h. The concentrations of various cytokines were measured in the cell culture supernatant after GO and V-rGO treatment (**A**–**F**). All values are in picograms and represented as the mean ± standard error (SE). * *P* < 0.05.

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
