# Peer review of "Differential Immunomodulatory Effect of Graphene Oxide and Vanillin-Functionalized Graphene Oxide Nanoparticles in Human Acute Monocytic Leukemia Cell Line (THP-1)"

_ijms, 2019, doi:10.3390/ijms20020247_

Round 1

Reviewer 1 Report

The manuscript by Gurunathan et al is an interesting work on the effect of GO and functionalazed GO with vanillin on differential immunomodulatory. My recommendation is to change reduce GO by functionalized GO as there are no clear evidences of the reduction from the point of view of the reviewer. It seems a classic functionalization.

Author Response

Response to the reviewer comments-1

We immensely thank the reviewer and the editor for their valuable and constructive comments that greatly facilitated us for improving the overall quality of the manuscript. As per the editor’s and reviewers’ constructive comments, the corrections were carried out in the manuscript. We hopefully believe that we have addressed all the comments mentioned by the reviewers carefully and precisely. All the changes are highlighted in yellow color in the revised manuscript. In addition, this manuscript was proof read by native English speaker by Editage editing company, Seoul, South Korea.

Comments and Suggestions for Authors

The manuscript by Gurunathan et al is an interesting work on the effect of GO and functionalazed GO with vanillin on differential immunomodulatory. My recommendation is to change reduce GO by functionalized GO as there are no clear evidences of the reduction from the point of view of the reviewer. It seems a classic functionalization.

First of all we are so thankful to the reviewer for encouraging, positive and constructive comments to improve overall quality of the manuscript. According to the reviewer comments, we have changed reduce GO into functionalized GO through the entire manuscript.

Once again thanks to the editor, reviewers for wonderful comments to improve the overall quality of the manuscript.

Reviewer 2 Report

Gurunathan et al. claim that GO and V-rGO stimulated the secretion of various cytokines and chemokines, indicating that the graphene-based materials induced potent inflammatory responses to THP-1 cells. This is an interesting study. However, there is a lot of unclear mechanism explanation in the strategy. For example, in this manuscript, there is no content related to the mechanism of cell death (in-vitro). Secondly, there is no comparison with state of the art reducing agents at different molecular levels. Based on the following concerns, I recommend this manuscript be accepted with subject to moderate revisions.

Following are the specific changes I can suggest;

-          Figure 1 E, F, would it be possible to remove the company name from the graph?

-          There are some typical functional groups and their spectral signatures (found in GO) which are missing in Figure 1 E such as C – O ( ≈ 1084 and ≈ 1387) and C – H ( ≈ 2921, ≈ 2849). Why are these functional groups are missing in GO prepared by following Hummer’s method?

-          What is effect of surface charge on (immuno)toxicity in a pH dependent manner? What is the pH of cells when used for toxicity screening?

-          Page 5, line 207 when authors say ‘further confirming the presence of a few layers of graphene sheets in V-rGO’. What are the number of layers? What does mean by few layers?

-          How the choice of concentration could be justified? The highest concentration is 100 µg/ml, which is still very low for real-world applications. How about the toxic impacts of GO/rGO if you increase the concentration like 500, 750 or 1000 µg/ml?

-          It is generally perceived that rGO is more toxic than GO. Although it depends on the reducing agent. Would it be possible for authors to explain around this? It would be useful for readers that how a reducing agent can make rGO toxic and how is this different in the case of this work?

-          How rGO can generate more ROS than GO when the functional groups have been reduced? EPR is the standard technique to measure ROS with/without cells. Would it be possible for authors to provide more evidence on ROS. Is there any role of singlet oxygen in (immune)toxicity? How about other oxygen-centred radicals?

-          Which species (ROS) did authors find in GO/rGO?

-          It has been well documented that cell line THP-1, a promyeloid cell line suggestive to outside factors, and hence sensitive to culture conditions. Can culture conditions alter THP-1 morphology and in turn affect their response to differentiation stimuli?

-          Since macrophages mediate innate immune responses and contribute to the adaptive immune response through their antigen presenting capacity it is relevant to understand their biology. THP-1 is the most common cell line utilized to study monocyte/macrophage differentiation and function. Did you see any effects on macrophages?

-          How were AgNPs were prepared? There is no detail on fabrication/characterization on these NPs.

Following reference can be added;

Yan, J., Chen, L., Huang, C. C., Lung, S. C. C., Yang, L., Wang, W. C., ... & Lin, C. H. (2017). Consecutive evaluation of graphene oxide and reduced graphene oxide nanoplatelets immunotoxicity on monocytes. Colloids and Surfaces B: Biointerfaces, 153, 300-309.

Wu, Y., Wang, F., Wang, S., Ma, J., Xu, M., Gao, M., & Liu, S. (2018). Reduction of graphene oxide alters its cyto-compatibility towards primary and immortalized macrophages. Nanoscale, 10(30), 14637-14650.

Tabish, T. A., Pranjol, M. Z. I., Hayat, H., Rahat, A. A., Abdullah, T. M., Whatmore, J. L., & Zhang, S. (2017). In vitro toxic effects of reduced graphene oxide nanosheets on lung cancer cells. Nanotechnology, 28(50), 504001.

Lategan, K., Alghadi, H., Bayati, M., de Cortalezzi, M. F., & Pool, E. (2018). Effects of Graphene Oxide Nanoparticles on the Immune System Biomarkers Produced by RAW 264.7 and Human Whole Blood Cell Cultures. Nanomaterials, 8(2), 125.

Cho, Y. C., Pak, P. J., Joo, Y. H., Lee, H. S., & Chung, N. (2016). In vitro and in vivo comparison of the immunotoxicity of single-and multi-layered graphene oxides with or without pluronic F-127. Scientific reports, 6, 38884.

Tabish, T. A., Zhang, S., & Winyard, P. G. (2017). Developing the next generation of graphene-based platforms for cancer therapeutics: the potential role of reactive oxygen species. Redox biology.

Zhi, X., Fang, H., Bao, C., Shen, G., Zhang, J., Wang, K., & Cui, D. (2013). The immunotoxicity of graphene oxides and the effect of PVP-coating. Biomaterials, 34(21), 5254-5261.

Author Response

Response to the reviewer comments-2

We immensely thank the reviewer and the editor for their valuable and constructive comments that greatly facilitated us for improving the overall quality of the manuscript. As per the editor’s and reviewers’ constructive comments, the corrections were carried out in the manuscript. We hopefully believe that we have addressed all the comments mentioned by the reviewers carefully and precisely. All the changes are highlighted in yellow color in the revised manuscript. In addition, this manuscript was proof read by native English speaker by Editage editing company, Seoul, South Korea.

Comments and Suggestions for Authors

Gurunathan et al. claim that GO and V-rGO stimulated the secretion of various cytokines and chemokines, indicating that the graphene-based materials induced potent inflammatory responses to THP-1 cells. This is an interesting study. However, there is a lot of unclear mechanism explanation in the strategy. For example, in this manuscript, there is no content related to the mechanism of cell death (in-vitro). Secondly, there is no comparison with state of the art reducing agents at different molecular levels. Based on the following concerns, I recommend this manuscript be accepted with subject to moderate revisions.

First of all we are so thankful to the reviewer for encouraging, positive and constructive comments to improve overall quality of the manuscript. According to the reviewer comments, we have addressed all the questions raised by the reviewer point by point.

Following are the specific changes I can suggest;

Figure 1 E, F, would it be possible to remove the company name from the graph? There are some typical functional groups and their spectral signatures (found in GO) which are missing in Figure 1 E such as C – O ( ≈ 1084 and ≈ 1387) and C – H ( ≈ 2921, ≈ 2849). Why are these functional groups are missing in GO prepared by following Hummer’s method?

Thanks to the reviewer for excellent comments. Response to your first question, we deleted the company name in the FTIR results. Response to your second question, we absolutely agree with reviewer comments. We also expected those typical functional groups and their spectral signatures, these functional groups are missing could be due to strong oxidation and sonication processes.  

What is effect of surface charge on (immuno)toxicity in a pH dependent manner? What is the pH of cells when used for toxicity screening?

Thanks to the reviewer for excellent question. The growing potential of graphene and graphene based nanomaterials in biomedical applications has provoked the urgent need to thoroughly address their interaction with biological systems. However, only limited studies have been performed to explore the effects of surface charge on the biological behaviors of graphene oxide and reduced graphene oxide. The zeta potential of prepared GO and reduced GO were 40 mV, and 20 mV and pH 7.0 of the cells were maintained through the entire experiment.

Page 5, line 207 when authors say ‘further confirming the presence of a few layers of graphene sheets in V-rGO’. What are the number of layers? What does mean by few layers?

Thanks to the reviewer for thought-provoking comments. Response to your first question, graphene is the common name to a single layer (mono-layer). Graphite is a three-dimensional material. Few-layers graphene is generally considered to be two (bi-layer), thee, and etc.

Response to your second question, few layer graphene means 2-3 layers and multi-layer graphene would go up to 5-10 layers.

How the choice of concentration could be justified? The highest concentration is 100 µg/ml, which is still very low for real-world applications. How about the toxic impacts of GO/rGO if you increase the concentration like 500, 750 or 1000 µg/ml?

Thanks to the reviewer for interesting comments. We performed dose-dependent toxicity up to 1000 µg/ml. The significant difference was observed from 20 to 100 µg/ml. While increasing concentration from 100 to 1000 µg/ml, there is no significant difference was observed between 100 and 1000 µg/ml.

It is generally perceived that rGO is more toxic than GO. Although it depends on the reducing agent. Would it be possible for authors to explain around this? It would be useful for readers that how a reducing agent can make rGO toxic and how is this different in the case of this work?

Thanks to the reviewer for excellent comment. The toxicity of GO and reduced GO is still controversy. These diverse transformation processes result in significant variations in the structural properties, reactivity, dispersibility, stability and density of functional groups. Furthermore reduced GO showed less affinity for extracellular proteins relative to the affinity shown by the parent GO. This reduced affinity consequently led to stronger interactions with the cell membrane and higher cellular uptake. Therefore, vanillin reduced GO increased cytotoxicity by causing more oxidative stress compared to parent GO.  Thus, these findings imply that although GO undergoes similar reduction processes, the difference in the reduction pathways causes striking variations in toxicity. Moreover, the great reactivity and stability of vanillin reduced graphene oxide produced graphene-based radicals might result in an even greater impact on toxicity.

How rGO can generate more ROS than GO when the functional groups have been reduced? EPR is the standard technique to measure ROS with/without cells. Would it be possible for authors to provide more evidence on ROS. Is there any role of singlet oxygen in (immune)toxicity? How about other oxygen-centred radicals?

Thanks to the reviewer for excellent question. Yes, we absolutely agree with reviewer comments. Response to your first question, we measured ROS using DCFH-DA, a fluorogenic dye that measures hydroxyl, peroxyl and other reactive oxygen species (ROS) activity within the cell. This assay can measure collective form reactive oxygen species. Graphene oxide is known to produce superoxide anion radical, singlet oxygen and hydroxyl radical and perhydroxyl radical and is termed collectively the “reactive oxygen species”. The graphene radical exhibited high intrinsic oxidative potential, which was confirmed by measuring the ability of the material to oxidize the florescence probe DCFH. The fluorescence intensity of rGO was almost two times those of GO. Carbon atoms in graphene are covalently bonded through three electrons from each atom to form a strong lattice, and the fourth valence electron on each carbon atom is delocalized. These half-filled orbitals of the carbon atoms form a bond that permits free motion of electrons. Therefore, these free electrons can be localized at the edge or defect sites of the graphene plane, which makes them more reactive and more likely to act as stable carbon based free radicals.

Response to your second question, our study is overall focused on the differential effect of GO and rGO on THP-1 cells not specifically focused on the role of singlet oxygen and oxygen centered free radicals in immune toxicity. However, the idea came from the reviewer is excellent idea; we will consider to do in future.

Which species (ROS) did authors find in GO/rGO?

Thanks to the reviewer for creative questions. Generally, GO and rGO produces singlet oxygen at the initial stage and then other reactive oxygen species including hydroxyl radicals are finally formed.

It has been well documented that cell line THP-1, a promyeloid cell line suggestive to outside factors, and hence sensitive to culture conditions. Can culture conditions alter THP-1 morphology and in turn affect their response to differentiation stimuli?

Thanks to reviewer for logical question. We carefully maintained the culture conditions for THP-1 cells. We didn’t observe any alteration in morphology.

Since macrophages mediate innate immune responses and contribute to the adaptive immune response through their antigen presenting capacity it is relevant to understand their biology. THP-1 is the most common cell line utilized to study monocyte/macrophage differentiation and function. Did you see any effects on macrophages?

Thanks to the reviewer for nice question. We didn’t observe any differentiation effect on macrophages.

How were AgNPs were prepared? There is no detail on fabrication/characterization on these NPs.

We included the methods for preparation and characterization of AgNPs in the revised manuscript.

Following reference can be added;

Yan, J., Chen, L., Huang, C. C., Lung, S. C. C., Yang, L., Wang, W. C., ... & Lin, C. H. (2017). Consecutive evaluation of graphene oxide and reduced graphene oxide nanoplatelets immunotoxicity on monocytes. Colloids and Surfaces B: Biointerfaces, 153, 300-309.

Wu, Y., Wang, F., Wang, S., Ma, J., Xu, M., Gao, M., & Liu, S. (2018). Reduction of graphene oxide alters its cyto-compatibility towards primary and immortalized macrophages. Nanoscale, 10(30), 14637-14650.

Tabish, T. A., Pranjol, M. Z. I., Hayat, H., Rahat, A. A., Abdullah, T. M., Whatmore, J. L., & Zhang, S. (2017). In vitro toxic effects of reduced graphene oxide nanosheets on lung cancer cells. Nanotechnology, 28(50), 504001.

Lategan, K., Alghadi, H., Bayati, M., de Cortalezzi, M. F., & Pool, E. (2018). Effects of Graphene Oxide Nanoparticles on the Immune System Biomarkers Produced by RAW 264.7 and Human Whole Blood Cell Cultures. Nanomaterials, 8(2), 125.

Cho, Y. C., Pak, P. J., Joo, Y. H., Lee, H. S., & Chung, N. (2016). In vitro and in vivo comparison of the immunotoxicity of single-and multi-layered graphene oxides with or without pluronic F-127. Scientific reports, 6, 38884.

Tabish, T. A., Zhang, S., & Winyard, P. G. (2017). Developing the next generation of graphene-based platforms for cancer therapeutics: the potential role of reactive oxygen species. Redox biology.

Zhi, X., Fang, H., Bao, C., Shen, G., Zhang, J., Wang, K., & Cui, D. (2013). The immunotoxicity of graphene oxides and the effect of PVP-coating. Biomaterials, 34(21), 5254-5261.

Thanks to the reviewer for indicating to include important references in the revised manuscript. According to the reviewer comments, we included all the references in the revised manuscript.

Once again thanks to the editor, reviewers for wonderful comments to improve the overall quality of the manuscript.